# A Comparative Study on Patient Safety Awareness Between Medical School Freshmen and Age-Matched Individuals

**DOI:** 10.3390/healthcare12222270

**Published:** 2024-11-14

**Authors:** Suguru Kohara, Kentaro Miura, Chie Sasamori, Shuho Hase, Akihito Shu, Kenji Kasai, Asuka Yokoshima, Naofumi Fujishiro, Yasuhiro Otaki

**Affiliations:** 1Department of Medicine, Teikyo University, Tokyo 173-8605, Japan; kenta.miura@jcom.zaq.ne.jp (K.M.);; 2Department of Obstetrics and Gynecology, St. Luke’s International Hospital, Tokyo 104-8560, Japan; 3Department of Pharmaceutical Sciences, Teikyo University, Tokyo 173-8605, Japan; 4Department of Neurosurgery, Kanto Medical Center NTT EC, Tokyo 141-8625, Japan; 5Department of Rehabilitation Medicine, Kameda Medical Center, Chiba 296-8602, Japan; 6General Medical Education and Research Center, Teikyo University, Tokyo 173-8605, Japan

**Keywords:** patient safety, medical school freshmen, medical students, pregraduate patient safety education, patient safety awareness

## Abstract

**Background**: To provide more effective pregraduate patient safety education, understanding medical students’ perceptions of patient safety before pregraduate patient safety education is necessary. Therefore, we conducted this study to examine patient safety awareness among medical students at the time of admission and compare it with that among controls. **Methods**: In the 2019 academic year, 132 medical school freshmen enrolled at Teikyo University and 166 age-matched, non-medical students enrolled at an affiliated institution within the Teikyo University organization were surveyed using an anonymous and self-administered questionnaire. The questionnaire divided patient safety awareness into three categories: perception, knowledge, and attitude, which were evaluated on a 5-point Likert scale (Cronbach’s alpha coefficient was 0.77). To assess overall patient safety awareness, the total scores were calculated for the item groups on “perception”, “knowledge”, and “attitude” and compared these scores between the two groups. **Results**: The total scores (mean ± SD) were 104.2 ± 10.2 for medical students and 88.8 ± 9.6 for controls (*p* < 0.001). In the “perception” and “attitude” item groups, a higher proportion of medical students provided a positive response than controls. In particular, medical students were more motivated to learn about patient safety than the controls. In the “knowledge” item group, neither medical students nor controls provided a high proportion of positive responses. **Conclusions**: Medical students demonstrated a higher awareness of patient safety than controls and showed a strong sensitivity to patient safety from the time of enrollment.

## 1. Introduction

In “To Err is Human: Building a Safer Health System” published by the United States Institute of Medicine in 1999, it was recorded that up to 98,000 hospitalized patients die annually due to “medical errors”, and the issue of patient safety attracted much attention [1]. In Japan during the same year, there was a patient mix-up at a university hospital and an inadvertent injection of disinfectant at a metropolitan hospital, leading to heightened public apprehension regarding patient safety [2]. Considering this movement, in 2002, the Japanese Ministry of Health, Labour and Welfare compiled “comprehensive measures for the promotion of medical safety” [3] to improve the environment in clinical settings and promote patient safety education. However, according to the 2021 annual report of the Japan Medical Safety Research Organization, which operates the medical accident investigation system, approximately 300 medical accidents were reported in the same year. The establishment of patient safety remains in the process of improvement [4]. Thus, patient safety is an important social issue worldwide, and patient safety education for healthcare professionals is ongoing [5,6,7,8].

The establishment of patient safety education in medical education institutions is equally important for improving patient safety in clinical practice [9,10,11,12,13,14]. In its 2011 Patient Safety Curriculum Guide, the World Health Organization urges the establishment of pregraduate patient safety education to further improve patient safety [15]. Similarly, in Japan, pregraduate patient safety education has attracted attention in recent years, with the introduction of the Model Core Curriculum for Medical Education in 2001 and a significant increase in the description of patient safety in the Model Core Curriculum for Medical Education in 2022 [16].

Since then, many studies measuring patient safety awareness among medical undergraduates have been conducted in various countries [17,18,19,20,21,22,23,24,25,26]. In Japan, Kasai et al. surveyed new medical undergraduate students and reported that a certain number of students have high patient safety awareness [27]. Furthermore, Shu et al. surveyed medical students at the time of admission and reported that a certain number of medical students have high patient safety awareness [28]. However, it remains unclear whether medical students have a higher level of patient safety awareness than controls. To determine how to provide better pregraduate patient safety education, understanding patient safety awareness among medical students at the time of admission compared to controls is necessary. Therefore, we conducted a study aimed to clarify the characteristics of patient safety awareness among medical school freshmen newly enrolled medical students of the Department of Medicine at Teikyo University (hereafter “medical students”) by comparing them with non-medical, age-matched students enrolled at an affiliated institution within the Teikyo University organization (hereafter “controls”), to discuss the introduction and timing of patient safety education in medical education.

## 2. Materials and Methods

### 2.1. Study Participants and Date of Study

The participants of this study were 132 medical school freshmen at the time of admission in the academic year of 2019, who had not yet received formal medical education. The controls of this study were 166 high school seniors enrolled at the affiliated institution, as they were closest in age and had a similar educational level within the Japanese educational system, in which medical schools accept students as early as the last term of their high school senior year. The survey was carried out in April 2019, shortly after the beginning of the new school year. The questionnaires were collected on the same day after all items had been answered.

### 2.2. Study Methods

An anonymous and self-administered questionnaire was used to collect data. The questionnaire used in this study was translated and modified from the questionnaire used in Nabilou et al.’s [10] study. The Cronbach’s alpha coefficient of the questionnaire used in this study was 0.77. The questionnaire asked about the students’ background and patient safety awareness, which were categorized into three groups: “perception”, “knowledge”, and “attitude”. Items 1–4 asked about the background of students between the two groups (i.e., sex, age, presence of medical professionals in the family, and whether the respondents themselves or someone close to them had experienced a medical accident). Items 5–15 asked about “perception”, items 16–25 asked about “knowledge”, and items 26–33 asked about “attitude”. Items 5–33 used a 5-point Likert scale (strongly agree/very good, agree/good, neutral/fair, disagree/poor, strongly disagree/very poor). In this study, positive responses were defined as “strongly agree” and “agree” in the affirmative form, whereas negative responses were defined as “disagree” and “strongly disagree” in the affirmative form. In the analysis, items in the negative form were reworded into the affirmative form. To assess overall patient safety awareness, the total scores were calculated for the item groups on “perception”, “knowledge”, and “attitude”, using a 5-point Likert scale—Strongly Agree (5 points), Agree (4 points), Neutral (3 points), Disagree (2 points), and Strongly Disagree (1 point)—and compared these scores between the two groups. The results are expressed as mean ± standard deviation.

In the development and validation of the questionnaire used in this study, experts in patient safety education as well as medical and pharmacy school students were involved. The questionnaire used by Nabilou et al. [10] was translated into Japanese and revised collaboratively by experts in patient safety education and medical English education while incorporating characteristics specific to patient safety education in Japan. We conducted a pilot study using the questionnaire on 20 medical school students in their second year and 20 pharmacy school students in their second year. Both groups had not received pregraduate patient safety education prior to the pilot study and did not participate in this study. Cronbach’s coefficient alpha test was used to analyze the results (α = 0.75) in accordance with the pilot study conducted by Nabilou et al. [10] (α = 0.72). The questionnaire used in this study primarily included questions related to objective elements, excluding ambiguous, subjective elements.

Before administering the questionnaires, the study participants were given a written description of the study, including the methods and objectives, with an additional verbal explanation. This description explained that participation was voluntary and that those who gave written consent were considered research participants. This study was approved by the Ethical Review Board for Medical and Health Research Involving Human Subjects of Teikyo University (authorization number: Teirin 17-104).

### 2.3. Statistical Analysis

The chi-square test was used to analyze attributes in items 1–4, while descriptive statistics were used to analyze responses to items regarding “perception”, “knowledge”, and “attitude” toward patient safety (items 5–33). Additionally, for each item from 5 to 33, statistical estimation was conducted. The total scores were calculated from student responses to each item based on a 5-point Likert scale—Strongly Agree (5 points), Agree (4 points), Neutral (3 points), Disagree (2 points), and Strongly Disagree (1 point)—and presented the mean values and 95% Confidence Intervals (CI) for the two groups. Cronbach’s coefficient alpha test was used to assess the reliability of the items. Student’s *t*-test was used to compare the total scores between medical students and controls. Kamran et al. [25] and Shu et al. [28], who assessed the level of patient safety awareness among medical school students, identified factors for patient safety awareness, including the presence of medical professionals in the family and the experience of medical accidents. Drawing on these studies, analysis of covariance (ANCOVA) was conducted to compare the level of patient safety awareness between the two groups, while controlling for the influence of sex, age, presence of medical professionals in the respondent’s family, and experiences of medical accidents of themselves or someone close to them. *p*-values of <0.05 were used to indicate statistical significance. R version 4.0.3 (R Core Team [2020], R: A language and environment for statistical computing, R Foundation for Statistical Computing, Vienna, Austria, http://www.R-project.org/ accessed on 25 July 2023) was used for the analysis.

## 3. Results

### 3.1. Response Rate and Respondent Demographics

Respondents who answered items 1–33 were defined as valid respondents. The demographics of the respondents are shown in Table 1. The response rate was 84.8% (112/132) for medical students and 86.7% (144/166) for controls. In regard to the distribution of sexes, there were 67 male medical students compared to 45 female students. Meanwhile, among controls, there were 66 male students and 78 female students (*p* = 0.026). The age of medical students was 20.3 ± 3.5 years, and that of controls was 17.0 ± 0.1 years (*p* < 0.001). Among medical students, 73.2% (82/112) had a healthcare provider in the family, while for controls, the percentage was 15.3% (22/144) (*p* < 0.001). In terms of reported medical accidents, 4.5% (5/112) of medical students and 4.9% (7/144) of controls indicated such experiences (*p* = 0.882).

### 3.2. “Perception”

Table 2 presents the responses of the items regarding “perception”. The proportion of students who provided positive responses for many items was higher for medical students than for controls. Medical students demonstrated a higher level of “perception” of patient safety compared to controls.

For item 8, “If I saw a medical error, I would report it to my supervisor”, which asked about the response when a medical accident occurs, 84.8% of medical students and 58.3% of controls provided positive responses. For item 10, “If a medical error occurs because of my medical practice, I would always report it to my supervisor”, which asked about the response upon causing a medical accident, 90.2% of the medical students and 71.6% of controls provided positive responses. Medical students were more inclined than controls to give a positive response to the item asking if you would report medical errors.

### 3.3. “Knowledge”

Table 3 shows the responses of the items regarding “knowledge”. The proportion of students who provided positive responses for many items was higher for medical students than for controls. Medical students demonstrated a higher level of “knowledge” regarding patient safety compared to controls.

For item 21, “You know about ‘team medicine’”, which assessed the knowledge of medical terminology related to patient safety, 93.8% of medical students and 28.4% of controls provided positive responses. For item 23, “You know about “informed consent”, which assessed the knowledge of medical terminology related to patient safety, 97.3% of medical students and 70.8% of controls provided positive responses. For item 17, “You know about “hiyari-hatto* (near-miss)”, which assessed the knowledge of medical terminology related to patient safety, 35.7% of medical students and 8.3% of controls provided positive responses. For item 18, “You know about “double-check”, which assessed the knowledge of medical terminology related to patient safety, 52.7% of medical students and 25.7% of controls provided positive responses. A comparison of medical students who provided positive and negative answers to items 17 and 18, which assessed the knowledge of practical and professional concepts, revealed similar levels of “perception” and “attitude” toward patient safety.

### 3.4. “Attitude”

Table 4 shows the results for the items regarding “attitude”. The proportion of students who provided positive responses for all items was higher for medical students than for controls. Medical students demonstrated a more positive “attitude” toward patient safety compared to controls.

For item 27, “‘Patient safety’ is an important topic in healthcare”, the proportion of students who provided positive responses was 99.1% for medical students and 93.1% for controls. For item 28, “Learning about patient safety is important in medical universities and colleges”, which asked about the importance of learning about patient safety in medical educational institutions, 98.2% of medical students and 88.9% of controls provided positive responses. For item 29, “You would like to learn more about ‘patient safety’”, which asked about the desire to learn about patient safety, 96.4% of medical students and 34.0% of controls provided positive responses. Additionally, we conducted an analysis focused on the total scores of the non-technical “attitude” items. The results indicated that medical students scored higher than controls, suggesting a greater awareness of patient safety.

Responses to the questionnaire were stratified according to sex, and responses to the “perception”, “knowledge”, and “attitude” categories of the questionnaires were analyzed. However, no major features were found that would affect our conclusions (Appendix A).

### 3.5. “Total Score”

The results for the total scores we calculated to assess overall patient safety awareness for the item groups on “perception”, “knowledge”, and “attitude” using a 5-point Likert scale are shown in Figure 1. The total scores were 104.2 ± 10.2 for medical students and 88.8 ± 9.6 for controls (mean difference: 15.5, *p* < 0.001, 95% CI: 13.0–17.9). The results of the covariates adjusted by analysis of covariance (ANCOVA) are shown in Table 5. Even after adjusting for confounding factors, a statistically significant difference in total scores between the two groups was observed (least square mean: 13.1, *p* < 0.001, 95% CI: 9.7–16.5). Medical students demonstrated a higher level of patient safety awareness compared to controls.

ANCOVA was also performed on the total scores, and a statistically significant difference was observed between the two groups (least squares mean: 13.1, *p* < 0.001, 95% CI: 9.7–16.5).

## 4. Discussion

This study aimed to clarify the characteristics of patient safety awareness among medical school freshmen at the time of admission, prior to formal medical education, by comparing them with controls. Additionally, we sought to discuss the introduction and timing of patient safety education in medical education. The results indicated that both medical students and controls demonstrated a high proportion of positive responses in the “perception” and “attitude” item groups, with medical students showing a significantly higher proportion of positive responses than controls. In the “knowledge” item group, both groups had relatively low proportions of positive responses. However, medical students still showed a higher proportion of positive responses compared to controls. Medical students also scored significantly higher than controls in total scores.

In this study, medical students had a higher awareness of reporting medical accidents than controls in the “perception” group (item 8 and item 10). Studies by Paterick et al. [29] and Aljabari et al. [30] revealed that medical professionals tend not to report medical errors because of fear of being held accountable. In contrast, according to Alshahrani et al. [11], Ezzi et al. [17], and Park et al. [19], medical students who received patient safety education are more aware of the need to report medical errors. Therefore, it is considered that adequate patient safety education, including the usefulness of medical incident reporting in patient safety, should be provided during medical school education. Medical students have been reported to be less motivated to learn with progression in medical school [31,32,33,34]. In this study, the “attitude” questions (items 27–29) revealed that medical students were significantly more motivated to learn about patient safety. Patient safety education from the time of admission, when students are significantly more motivated to learn, would have a higher educational impact than that during the subsequent years of school.

Studies by Nabilou et al. [10] and Svitlica et al. [35] reported that even medical students who received patient safety education had inadequate knowledge regarding patient safety. In the “knowledge” questions in our study, medical students demonstrated a solid understanding of broadly recognized concepts, such as team medicine (item 21) and informed consent (item 23), but showed limited knowledge of practical and specialized concepts like hiyari-hatto (near-miss) (item 17) and double-check (item 18), likely due to their lack of formal medical education. However, medical students displayed a higher level of specialized knowledge compared to controls. This suggests that medical school freshmen may have more knowledge about patient safety—even in areas they may not find particularly interesting—compared to non-medical students, suggesting a broader interest in various aspects of medicine. The analysis of items 17 and 18, along with the “perception” and “attitude” item groups, reveals that medical students have a high motivation to learn about patient safety, regardless of their current level of knowledge. Considering previous studies reporting declines in motivation as academic years progress [31,32,33,34], these findings underscore the importance of implementing pre-graduate patient safety education early, covering both foundational and specialized knowledge.

The results of this study suggested that medical school freshmen exhibit sensitivity not only to the technical aspects of medicine but also to patient safety from the outset of their medical education, indicating that their professional interests may already be well-formed in various areas. Furthermore, our findings implied that these freshmen are likely to actively engage in patient safety initiatives during their university education. Given their heightened awareness of patient safety, leveraging and enhancing this awareness could lead to more effective patient safety education. Currently, in many Japanese universities, patient safety education is introduced in the third or fourth year of medical school education, which lasts for 6 years. As previous studies have shown that patient safety awareness increases following the introduction of patient safety education [22,35], the results in this study suggested that it would be beneficial to implement a patient safety education curriculum from an earlier stage in medical education, incorporating more specialized contents.

### Limitations and Future Directions of the Study

This study has certain limitations. This study is a cross-sectional study comparing 132 medical school freshmen enrolled at Teikyo University in 2019 with 166 age-matched, non-medical students enrolled at the affiliated institution. Further validation is necessary by including medical school freshmen from other universities in Japan as well as students from other countries with diverse educational and cultural backgrounds. Additionally, studies conducted across different years into medical school education would strengthen these findings.

While this study suggested the effectiveness of earlier patient safety education, further research could confirm these results by comparing patient safety awareness post-education or at graduation between cohorts receiving early versus current timing of patient safety training.

## 5. Conclusions

In Japan, approximately 300 medical accidents were reported to the Medical Accident Investigation System annually, indicating the importance of patient safety education. However, patient safety education curricula widely vary among medical schools and have not yet been standardized across the country. This study aimed to clarify the characteristics of patient safety awareness among medical school freshmen prior to receiving patient safety education by comparing them with age-matched individuals who are not pursuing a medical career. We also aimed to discuss the introduction of patient safety education in medical schools and the optimal timing for its implementation.

Our findings revealed that medical school freshmen demonstrated significantly higher motivation to learn about patient safety compared to their age-matched controls. This suggested that medical school freshmen are interested not only in the technical aspects of medicine but also in patient safety. In other words, medical school freshmen appear to have already developed professional interests at the time of admission, encompassing not only technical aspects of healthcare but also broader areas such as patient safety. Therefore, incorporating continuous patient safety education from the first year—when students’ motivation to learn is considered to be at its highest—may enhance medical students’ and, ultimately, physicians’ awareness of patient safety, contributing to the promotion of safer healthcare practices.

## Figures and Tables

**Figure 1 healthcare-12-02270-f001:**
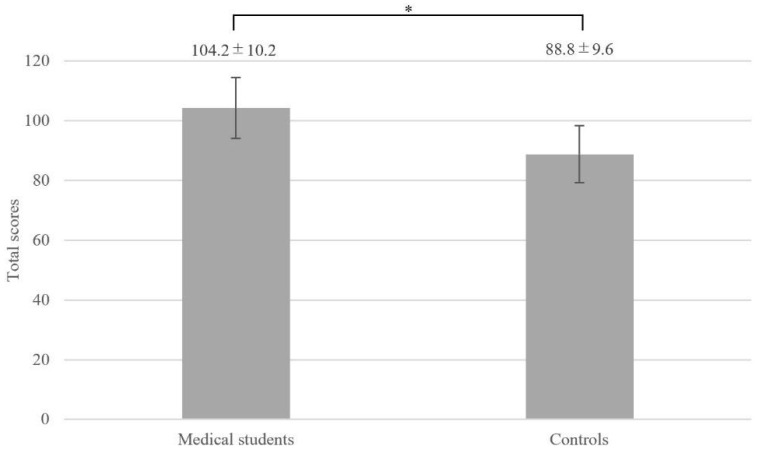
The results for the total scores of patient safety awareness. The total scores were 104.2 ± 10.2 for medical students and 88.8 ± 9.6 for controls, respectively. * (mean difference: 15.5, *p* < 0.001, 95% CI: 13.0–17.9).

**Table 1 healthcare-12-02270-t001:** Demographics of the respondents.

Items 1–4 (Respondent Demographics)		Medical Students	Controls	*p-*Value
Sex	Male	67	66	0.026
	Female	45	78	
Age	≤18	23	144	<0.001
	≥19	89	0	
Presence of medical professionals in the family	Yes	82	22	<0.001
	No	30	122	
Experience of medical accidents	Yes	5	7	0.882
	No	107	137	

**Table 2 healthcare-12-02270-t002:** Student responses to “perception” items regarding patient safety awareness *.

“Perception” Items		Strongly Agree (%)	Agree (%)	Neutral (%)	Dis Agree (%)	Strongly Disagree (%)	Mean Value of Total Points (95% CI)
5 Medical errors are inevitable	Medical students	15.2	60.7	10.7	10.7	2.7	2.3 (2.1–2.4)
	Controls	9.7	57.6	23.6	7.6	1.4	2.3 (2.2–2.5)
6 Competent physicians do not make medical errors that lead to patient harm	Medical students	1.8	10.7	20.5	45.5	21.4	3.7 (3.6–3.9)
	Controls	4.9	16.0	19.4	50.0	9.7	3.4 (3.3–3.6)
7 Medical errors can be eliminated through the efforts of physicians	Medical students	0.9	41.1	25.9	25.9	6.3	3.0 (2.9–3.2)
	Controls	7.6	34.7	31.9	22.9	2.8	3.2 (3.1–3.4)
8 If I saw a medical error, I would report it to my supervisor	Medical students	33.9	50.9	9.8	4.5	0.9	4.1 (4.0–4.3)
	Controls	13.2	45.1	24.3	15.3	2.1	3.5 (3.4–3.7)
9 If there is no harm to a patient, then there is no need to report medical errors to my supervisor	Medical students	0.0	8.9	12.5	37.5	41.1	4.1 (3.9–4.3)
	Controls	2.1	9.0	11.8	45.8	31.3	4.0 (3.8–4.1)
10 If a medical error occurs because of my medical practice, I would always report it to my supervisor	Medical students	43.8	46.4	7.1	2.7	0.0	4.3 (4.2–4.4)
	Controls	18.8	52.8	19.4	8.3	0.7	3.8 (3.7–3.9)
11 Establishing a system for reporting medical errors will lead to a reduction in the number of such errors	Medical students	36.6	44.6	8.9	9.8	0.0	4.1 (3.9–4.3)
	Controls	16.7	51.4	17.4	12.5	2.1	3.7 (3.5–3.8)
12 Working more carefully can effectively prevent the recurrence of similar medical errors	Medical students	29.5	57.1	5.4	8.0	0.0	4.1 (3.9–4.2)
	Controls	25.0	62.5	8.3	4.2	0.0	4.1 (4.0–4.2)
13 Punishing the parties involved in medical errors does not reduce medical errors	Medical students	8.9	40.2	29.5	18.8	2.7	2.7 (2.5–2.8)
	Controls	13.2	36.8	29.9	18.1	2.1	2.6 (2.4–2.8)
14 Increased safety awareness within hospitals would help reduce medical errors	Medical students	38.4	58.9	2.7	0.0	0.0	4.4 (4.3–4.5)
	Controls	29.9	59.0	9.0	2.1	0.0	4.2 (4.1–4.3)
15 Healthcare professionals actively report medical errors to reduce the number of such errors	Medical students	1.8	24.1	44.6	27.7	1.8	3.0 (2.8–3.1)
	Controls	2.8	10.4	45.8	35.4	5.6	2.7 (2.6–2.8)

* The total scores were calculated from student responses to each item based on a 5-point Likert scale—Strongly Agree (5 points), Agree (4 points), Neutral (3 points), Disagree (2 points), and Strongly Disagree (1 point)—and presented the mean values and 95% CI for the two groups.

**Table 3 healthcare-12-02270-t003:** Student responses to “knowledge” items regarding patient safety awareness.

“Knowledge” Items		Very Good (%)	Good (%)	Fair (%)	Poor (%)	Very Poor (%)	Mean Value of Total Points (95% CI)
16 You know about “time out”	Medical students	0.0	7.1	2.7	30.4	59.8	1.6 (1.4–1.7)
	Controls	4.2	7.6	6.9	15.3	66.0	1.7 (1.5–1.9)
17 You know about “hiyari-hatto *”	Medical students	14.3	21.4	2.7	19.6	42.0	2.5 (2.2–2.8)
	Controls	0.7	7.6	4.9	7.6	79.2	1.4 (1.3–1.6)
18 You know about “double-check”	Medical students	9.8	42.9	4.5	12.5	30.4	2.9 (2.6–3.2)
	Controls	4.2	21.5	11.1	12.5	50.7	2.2 (1.9–2.4)
19 You know about the “Medical Accident Investigation System ^†^”	Medical students	4.5	18.8	2.7	25.9	48.2	2.1 (1.8–2.3)
	Controls	0.7	5.6	9.0	18.1	66.7	1.6 (1.4–1.7)
20 You know about “triage”	Medical students	18.8	37.5	0.9	15.2	27.7	3.0 (2.8–3.3)
	Controls	9.0	12.5	5.6	6.3	66.7	1.9 (1.7–2.1)
21 You know about “team medicine”	Medical students	27.7	66.1	5.4	0.0	0.9	4.2 (4.1–4.3)
	Controls	8.3	20.1	9.0	18.8	43.8	2.3 (2.1–2.5)
22 You are aware of an accident in which several patients died after undergoing laparoscopic surgery at a university hospital ^‡^	Medical students	14.3	42.9	1.8	21.4	19.6	3.1 (2.8–3.4)
	Controls	2.1	26.4	4.9	16.7	50.0	2.1 (1.9–2.4)
23 You know about “informed consent”	Medical students	32.1	65.2	1.8	0.0	0.9	4.3 (4.2–4.4)
	Controls	13.9	56.9	6.9	5.6	16.7	3.5 (3.2–3.7)
24 You know about “evidence-based medicine”	Medical students	11.6	43.8	4.5	22.3	17.9	3.1 (2.8–3.3)
	Controls	1.4	6.9	7.6	14.6	69.4	1.6 (1.4–1.7)
25 Communication skills of healthcare professionals are relevant to medical errors	Medical students	18.8	54.5	12.5	10.7	3.6	3.7 (3.6–3.9)
	Controls	4.2	22.2	13.9	24.3	35.4	2.4 (2.1–2.6)

* This term is “near-miss” in Japanese. † The Medical Accident Investigation System was implemented by the Japan Medical Safety Research Organization (Medsafe Japan) in October 2015. This system targets unforeseen death caused by medical care that was reported as a medical accident, defined as “death or stillbirth cases that are caused or may have been caused by the care provided by employees of the medical institutions and are unforeseen by the administrator”. This system aims to promote medical safety by preventing the recurrence of similar medical accidents. ‡ Medical accidents at Gunma University Hospital occurred from 2010 to 2014 that resulted in the death of eight patients after laparoscopic surgery by the same doctors.

**Table 4 healthcare-12-02270-t004:** Student responses to “attitude” items regarding patient safety awareness.

“Attitude” Items		Strongly Agree (%)	Agree (%)	Neutral (%)	Disagree (%)	Strongly Disagree (%)	Mean Value of Total Points (95% CI)
26 Healthcare professionals should routinely spend part of their professional time in improving patient care	Medical students	60.7	38.4	0.9	0.0	0.0	4.6 (4.5–4.7)
	Controls	38.2	52.1	6.9	0.7	2.1	4.2 (4.1–4.4)
27 “Patient safety” is an important topic in healthcare	Medical students	67.0	32.1	0.9	0.0	0.0	4.7 (4.6–4.8)
	Controls	36.8	56.3	5.6	0.0	1.4	4.3 (4.2–4.4)
28 Learning about patient safety is important in medical universities and colleges	Medical students	63.4	34.8	1.8	0.0	0.0	4.6 (4.5–4.7)
	Controls	41.0	47.9	9.7	0.0	1.4	4.3 (4.1–4.4)
29 You would like to learn more about “patient safety”	Medical students	44.6	51.8	2.7	0.9	0.0	4.4 (4.3–4.5)
	Controls	8.3	25.7	38.2	15.3	12.5	3.0 (2.8–3.2)
30 You do not wish to support or advise a peer to decide how to respond to a medical error	Medical students	1.8	3.6	17.0	50.9	26.8	4.0 (3.8–4.1)
	Controls	2.8	6.3	34.0	38.2	18.8	3.6 (3.5–3.8)
31 You want to analyze a case to find the cause of a medical error	Medical students	47.3	50.9	1.8	0.0	0.0	4.5 (4.4–4.6)
	Controls	33.3	54.9	11.1	0.0	0.7	4.2 (4.1–4.3)
32 You will not disclose a medical error to the patient	Medical students	2.7	19.6	33.9	33.0	10.7	3.3 (3.1–3.5)
	Controls	4.2	22.9	37.5	22.2	13.2	3.2 (3.0–3.3)
33 You will share all facts of the medical error with your colleagues to prevent recurrence	Medical students	26.8	53.6	17.0	2.7	0.0	4.0 (3.9–4.2)
	Controls	22.2	49.3	25.0	2.8	0.7	3.9 (3.8–4.0)

**Table 5 healthcare-12-02270-t005:** The results of the covariates adjusted by ANCOVA.

	Least Square Mean	95% Confidence Interval	*p-*Value
Medical students/Controls	13.1	9.7–16.5	<0.001
Sex (Male/Female)	−3.8	−6.2 to −1.4	0.002
Age (≦18/19≦)	−0.8	−1.3 to −0.2	0.006
Presence of medical professionals in the family (Yes/No)	0.6	−2.3 to 3.6	0.673
Experience of medical accidents (Yes/No)	2.5	−3.4 to 8.5	0.402

## Data Availability

The original contributions presented in the study are included in the article. Further inquiries can be directed to the corresponding author.

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
