# Peer review of "A Comparative Study on Patient Safety Awareness Between Medical School Freshmen and Age-Matched Individuals"

_healthcare, 2024, doi:10.3390/healthcare12222270_

Round 1
Reviewer 1 Report
Comments and Suggestions for Authors
Kohara et al, tried to evaluate patient safety awareness among medical school freshmen and high school seniors. The subject is very interesting, however, the application seems to be inadequate and the objective of the study is not clear.
In fact, the authors compared between medical and non medical students which is not really adequate in this case.
Comparing medical and non medical subject for a subject purely medical seems to be misleading.
Some items are not adapted for non medical students (i.e: If I saw a medical error, I would report it to my supervisor, If there is no harm to a patient, then there is no need to report medical errors to my supervisor, item 10, item 11, 28, 29…).
Other items are very technical and cannot be understood by non medical students (questions 16-25)
Author Response
Dear Reviewer 1,
Thank you very much for your valuable feedback. We deeply appreciate your insightful comments.
1)Reviewer 1 commented “Kohara et al, tried to evaluate patient safety awareness among medical school freshmen and high school seniors. The subject is very interesting, however, the application seems to be inadequate and the objective of the study is not clear.
In fact, the authors compared between medical and non medical students which is not really adequate in this case.
Comparing medical and non medical subject for a subject purely medical seems to be misleading.
Some items are not adapted for non medical students (i.e: If I saw a medical error, I would report it to my supervisor, If there is no harm to a patient, then there is no need to report medical errors to my supervisor, item 10, item 11, 28, 29…).
Other items are very technical and cannot be understood by non medical students (questions 16-25)”
Thank you very much for your valuable feedback.
First, I'd like to provide some background on the Japanese education system. In Japan, most medical students enter medical school directly after high school. Therefore, we chose high school seniors as the comparison group in this study to minimize the age gap. However, since some students may spend a few years preparing for the medical school entrance exam, there can be age differences among incoming students. To account for this, we have adjusted for age using ANCOVA to mitigate any potential age-related effects.
The participants in this study are medical school freshmen who have just begun their medical education. The only difference between them and our comparison group—third-year high school students—is whether or not they have applied to medical school. The questionnaire used to assess patient safety awareness includes medical content in the "knowledge" section, while the "perception" and "attitude" sections address more general topics. Since the goal of this study is to compare the two groups rather than assess specific medical comprehension, this questionnaire is well-suited for both medical school freshmen and high school seniors.
In response to your comment regarding the appropriateness of comparing medical students with non-medical students, we emphasize that this study focuses on attitudes in both groups prior to university-level education. Certain items (e.g., items 10, 11, 28, 29) assess fundamental patient safety attitudes that both groups can reasonably respond to without formal training. Additionally, for items noted as "unintelligible" (e.g., items 16-25), we clarify that these were included to establish baseline knowledge differences rather than test comprehension.
The aim of this study is to identify patient safety awareness characteristics in medical school freshmen upon admission and to assess the optimal timing for introducing patient safety education. This objective is clearly stated in the revised manuscript’s Introduction, on page 2, line 72 to 77.

Reviewer 2 Report
Comments and Suggestions for Authors
A Comparative Study of Patient Safety Awareness Among Medical School Freshmen and High School Seniors
Thank you for the opportunity to review the manuscript and for your work in medical education and promoting patient safety among Japanese medical students. While I think the manuscript has some potential, I think it is lacking in a number of areas, that I have highlighted below. My concerns are not so much with the data, but rather how the data are presented and interpreted. Below I offer my reflections and hope that the authors find them constructive.
Major:
1. There’s some confusion in the paper regarding the description of the groups that you are comparing. In your title you emphasize “Medical School Freshmen and High School Seniors”. However, in your abstract and introduction you make the point that the high school students represent members of “the general public of the same age group”. It would have made more sense to survey medical students versus non-medical students. This issue is further emphasized when in the results you mention age and how there’s a significant difference, namely, significantly more high school seniors are aged 18. This finding is not only obvious, but your emphasis on this point seems to detract from your assertion that the high school seniors represent members of the general public of the same age group. Maybe you need to refer to your control group as something other than “high school seniors” which would perhaps mean changing the title and the name of the group throughout the text? Or you could lean into the fact that you are surveying two cohorts who are of a similar age, but at two different stages of their educational journey, however, in that case it would have been better to survey freshmen medical students versus high school seniors planning to enter medical school.
2. My second major concern is the Discussion section, which I think severely lacks any substantive argument from your data. The first paragraph of the Discussion section is simply results. Please interpret what your findings indicate.
These two sections essentially say the same thing except that attitude and knowledge categories are different:
“Conversely, medical students have been reported to be less motivated to learn with progression in medical school [31]. In this study, the “attitude” questions (Q27–Q29) revealed that medical students were significantly more motivated to learn about patient safety. Patient safety education from the time of admission, when students are significantly more motivated to learn, would have a higher educational impact than that during the subsequent years of school.”
“In consideration of previous studies revealing a decrease in motivation to learn with progression in medical school [31], it is considered that sufficient pregraduate patient safety education, including knowledge of patient safety, should be provided from the time of admission, when students have a significantly high motivation to learn, even in terms of “knowledge.”
“Therefore, it is considered that pregraduate patient safety education from the time of enrollment is necessary to foster patient safety awareness among medical students.”
This sentence restates the same point.
“Therefore, it is considered that pregraduate patient safety education from the time of enrollment is necessary to foster patient safety awareness among medical students.”
Is the only conclusion to be drawn from your study is that patient safety should be taught in the first year of medical school?
Additionally, you base your suggestion that patient safety should be taught in the first year of medical school on the assertion that motivation among medical students dwindles as they progress through the years, but you only cite one study that makes this point (doi: 10.5116/ijme.565e.0f87). This study involves a cohort of only 43 students at a single institution in South Korea and I don’t think this is sufficient to make the argument nor true of every medical school. How about Japan? How about globally? What is the reason for the decrease in motivation?
Again, while I think your I think your data is adequate, I think you should think more about the results and implications of your findings. For example, can you state what aspect of patient safety should be taught? What particular knowledge is lacking among your cohort? Why? Is there anything particular about the Japanese context of your study that reveals any culturally specific trends? Etc. how does “perception”, “knowledge”, and “attitude” differ by sex.
3. My third concern or rather question is that you include sex in the demographic information, yet you do not offer any observation regarding variations in your results by sex. It would be interesting, for example, to know if female or male medical students had more knowledge, awareness, etc. of patient safety, or what aspects males and females scored higher on, and so on. I think if you could present this data and then speculate as to why there was some difference (if any), I think it would add an important dimension to the paper.
4. Finally, in the abstract you state that: “Medical students at the time of admission have higher patient safety awareness than high school seniors.” This finding is too obvious that it makes me question why you did the study. This relates to my first point above about the naming of the groups, but also a conclusion calls for something a bit more novel.
Minor comments:
1. I think you should always define the term “hiyari-hatto” in English in parentheses at every mention in the text.
2. Please add: “the United States’” before “Institute of Medicine” and “the Japanese” before “Ministry of Health” so that the reader knows the context of each statement better.
3. “to ensure medical safety” change the word “ensure” to “promote.” It is impossible to “ensure” patient safety.
4. For style, please be aware of the repetition of words as in this part: “patient safety remains in the process of improvement [4]. Thus, improving the quality of healthcare and establishing patient safety remain….”
5. For style, please be aware of redundancy. For example, here you mention “patient safety” three times in one sentence: “Similarly, in Japan, pregraduate patient safety education has attracted attention in recent years, with the introduction of pregraduate patient safety education in the Model Core Curriculum for Medical Education in 2001 and a significant increase in the description of patient safety in the Model Core Curriculum for Medical Education in 2022.”
6. Please be aware of the verb past tense in the sentences. For example, in the introduction: “and compare it with” should be in the past tense.
7. Some of the sentences are awkward or lack logical flow. For example: “Medical students were provided survey forms in April 4th, 2019, immediately after their enrollment. Furthermore, third-year high school students were provided the forms in April 9th, 2019, immediately after progressing to the third year. The questionnaires were collected on the same day after all questions were answered.” How about something like: “The survey was carried out in April 2019, shortly after beginning the new school year.”
7. In section 2.2 you mention: “The Cronbach’s alpha coefficient was 0.77.” However, at this stage it is not clear at this point if the is the Cronbach’s alpha for Nabilou et al.’s survey or for your Japanese version. Please clarify for the reader.
8. Here I think it would be good to include (in parentheses) information regarding the meanings of the terms, or content of the questions as you did in the preceding sentence regarding background: “Questions 5–15 asked about “perception,” questions 16–25 asked about “knowledge,” and questions 26–33 asked about “attitude.”
9. Please be aware of the changes of voice, for example in the Materials and Methods section: “We calculated the total scores” should be in the passive voice to match the rest of that section.
10. In the results section, you refer to the items in the questionnaire as “questions”; however, they are written as statements. Maybe it would be better to refer to them as “items.” I think you can also remove the word “Question” from the tables and just have the numbers.
11. Please use the word “participants” rather than “subjects”
Comments on the Quality of English Language
The English need some revision, preferably by an experienced native English speaker editor.
Author Response
Dear Reviewer 2,
Thank you very much for your valuable feedback. We deeply appreciate your insightful comments.
1)Reviewer 2 commented “There’s some confusion in the paper regarding the description of the groups that you are comparing. In your title you emphasize “Medical School Freshmen and High School Seniors”. However, in your abstract and introduction you make the point that the high school students represent members of “the general public of the same age group”. It would have made more sense to survey medical students versus non-medical students. This issue is further emphasized when in the results you mention age and how there’s a significant difference, namely, significantly more high school seniors are aged 18. This finding is not only obvious, but your emphasis on this point seems to detract from your assertion that the high school seniors represent members of the general public of the same age group. Maybe you need to refer to your control group as something other than “high school seniors” which would perhaps mean changing the title and the name of the group throughout the text? Or you could lean into the fact that you are surveying two cohorts who are of a similar age, but at two different stages of their educational journey, however, in that case it would have been better to survey freshmen medical students versus high school seniors planning to enter medical school.”
Thank you for your valuable feedback.
In Japan, it is common for students to enter medical school directly after high school graduation. Therefore, as a comparison group in this study, we selected third-year high school students who are close in age and do not aspire to enter medical school. Since medical school admission is highly competitive, there may be age differences among incoming students. We used ANCOVA to appropriately adjust for the impact of these age differences.
The participants in this study are medical school freshmen who have not yet begun formal medical education; therefore, the primary difference between them and the high school comparison group is the intent to pursue medicine, with both groups at a similar educational stage. In line with your suggestions, we have changed the name of the control group to “general public of the same age” and updated the references throughout the text to “general public.”
We plan to conduct a future study comparing medical school freshmen with non-medical university students to further validate our findings.
Based on your suggestions, we have incorporated these additions into the revised manuscript’s Materials and Methods Section, with adjustments on pages 2, line 83 to line 92.
2) Reviewer 2 commented “My second major concern is the Discussion section, which I think severely lacks any substantive argument from your data. The first paragraph of the Discussion section is simply results. Please interpret what your findings indicate.
These two sections essentially say the same thing except that attitude and knowledge categories are different:
“Conversely, medical students have been reported to be less motivated to learn with progression in medical school [31]. In this study, the “attitude” questions (Q27–Q29) revealed that medical students were significantly more motivated to learn about patient safety. Patient safety education from the time of admission, when students are significantly more motivated to learn, would have a higher educational impact than that during the subsequent years of school.”
“In consideration of previous studies revealing a decrease in motivation to learn with progression in medical school [31], it is considered that sufficient pregraduate patient safety education, including knowledge of patient safety, should be provided from the time of admission, when students have a significantly high motivation to learn, even in terms of “knowledge.”
“Therefore, it is considered that pregraduate patient safety education from the time of enrollment is necessary to foster patient safety awareness among medical students.”
This sentence restates the same point.
“Therefore, it is considered that pregraduate patient safety education from the time of enrollment is necessary to foster patient safety awareness among medical students.”
Is the only conclusion to be drawn from your study is that patient safety should be taught in the first year of medical school?
Additionally, you base your suggestion that patient safety should be taught in the first year of medical school on the assertion that motivation among medical students dwindles as they progress through the years, but you only cite one study that makes this point (doi: 10.5116/ijme.565e.0f87). This study involves a cohort of only 43 students at a single institution in South Korea and I don’t think this is sufficient to make the argument nor true of every medical school. How about Japan? How about globally? What is the reason for the decrease in motivation?
Again, while I think your I think your data is adequate, I think you should think more about the results and implications of your findings. For example, can you state what aspect of patient safety should be taught? What particular knowledge is lacking among your cohort? Why? Is there anything particular about the Japanese context of your study that reveals any culturally specific trends? Etc. how does “perception”, “knowledge”, and “attitude” differ by sex.”
Thank you for your valuable feedback.
We have restructured the Discussion Section. Medical school freshmen demonstrated a high level of interest in and motivation to learn about patient safety. They possessed basic knowledge of patient safety; however, as they had not yet begun formal medical education, they lacked understanding of more specialized concepts, such as “hiyari-hatto (near-miss)”(item 17) and double-check (item 18). Based on these findings, it appears beneficial to implement pre-graduation patient safety education early in their training, covering both foundational and specialized knowledge.
These results suggest that medical school freshmen already show sensitivity not only to the technical aspects of medicine but also to patient safety, indicating that their professional interests may already be well-formed in various areas. The study further implies that these freshmen are likely to engage actively in patient safety topics during their university education. Given their heightened awareness of patient safety, leveraging and strengthening this orientation could lead to more effective patient safety education.
In Japan, patient safety education is typically introduced in the third or fourth year of medical school. However, since studies have reported that medical students’ motivation tends to decrease as they progress through school in Japan and other countries [1-4], our findings indicate that a curriculum that introduces patient safety education earlier and with more specialized content would be advantageous. Based on your suggestions, we have incorporated these additions into the revised manuscript’s Discussion Section, with adjustments on pages 8, line 259 to page 9, line 310.
References
- Nakashima et al. Surveys to assess the attitudes of medical students about learning. igakukiyouiku 2010,41(6):429-434. doi: https://doi.org/10.11307/mededjapan.41.429.
- Wu H, Li S, Zheng J, Guo J. Medical students' motivation and academic performance: the mediating roles of self-efficacy and learning engagement. Med Educ Online. 2020 Dec;25(1):1742964. doi: 10.1080/10872981.2020.1742964. PMID: 32180537; PMCID: PMC7144307.
- Sarkis AS, Hallit S, Hajj A, Kechichian A, Karam Sarkis D, Sarkis A, Nasser Ayoub E. Lebanese students' motivation in medical school: does it change throughout the years? A cross-sectional study. BMC Med Educ. 2020 Mar 31;20(1):94. doi: 10.1186/s12909-020-02011-w. PMID: 32234030; PMCID: PMC7110720.
- Kim KJ, Jang HW. Changes in medical students' motivation and self-regulated learning: a preliminary study. Int J Med Educ. 2015 Dec 28;6:213-5. doi: 10.5116/ijme.565e.0f87. PMID: 26708325; PMCID: PMC4695391.
3)Reviewer 2 commented “ My third concern or rather question is that you include sex in the demographic information, yet you do not offer any observation regarding variations in your results by sex. It would be interesting, for example, to know if female or male medical students had more knowledge, awareness, etc. of patient safety, or what aspects males and females scored higher on, and so on. I think if you could present this data and then speculate as to why there was some difference (if any), I think it would add an important dimension to the paper.”
Thank you for your valuable feedback.
We conducted a gender-specific analysis, which revealed no significant differences based on gender, and thus, this did not impact the conclusions of our study. The gender-specific data has been included in the Supplement (Table S1-S3). In this study, ANCOVA was used to appropriately adjust for the potential influence of gender, age, presence of healthcare professionals in the family, and prior experience with medical accidents. These additions have been incorporated into the revised manuscript’s Results Section, with adjustments on page 7, line 231 to 234.
4)Reviewer 2 commented “Finally, in the abstract you state that: “Medical students at the time of admission have higher patient safety awareness than high school seniors.” This finding is too obvious that it makes me question why you did the study. This relates to my first point above about the naming of the groups, but also a conclusion calls for something a bit more novel.”
Thank you for your valuable feedback.
We have revised the summary conclusion to “Medical students demonstrated a higher awareness of patient safety than the general public and showed a strong sensitivity to patient safety from the time of enrollment.” This change has been incorporated into the revised manuscript’s Abstract Section, with adjustments on pages 1, lines 34 to 36. Additionally, based on the suggestions for the Abstract Section, we have added a Conclusion Section and improved the text on page 9, line 322 to page 10, line 328 of the revised manuscript.
Minor comments:
- Reviewer 2 commented “I think you should always define the term “hiyari-hatto” in English in parentheses at every mention in the text”.
Thank you for your valuable feedback.
We have revised the term "hiyari-hatto" to "hiyari-hatto (near-miss)" throughout the manuscript. These changes have been incorporated into the revised manuscript’s Results and Discussion Sections, with adjustments on page 6, line 206 and page 9, line 291.
- Reviewer 2 commented “Please add: “the United States’” before “Institute of Medicine” and “the Japanese” before “Ministry of Health” so that the reader knows the context of each statement better”.
Thank you for your valuable feedback.
We have revised the terms to “the United States Institute of Medicine” and “the Japanese Ministry of Health,” throughout the manuscript. These changes have been incorporated into the revised manuscript’s Introduction Section, with adjustments on page 1, line 41 and page 2, line 47.
- Reviewer 2 commented ““to ensure medical safety” change the word “ensure” to “promote” It is impossible to “ensure” patient safety”.
Thank you for your valuable feedback.
We have revised the wording to “to promote medical safety” throughout the manuscript. This change has been incorporated into the revised manuscript’s Introduction Section, with adjustments on page 2, line 48.
- Reviewer 2 commented “For style, please be aware of the repetition of words as in this part: “patient safety remains in the process of improvement [4]. Thus, improving the quality of healthcare and establishing patientsafety remain….”
Thank you for your valuable feedback.
We have made the suggested revisions. These changes have been incorporated into the revised manuscript’s Introduction Section, with adjustments on page 2, line 52 to 54.
- Reviewer 2 commented “For style, please be aware of redundancy. For example, here you mention “patient safety” three times in one sentence: “Similarly, in Japan, pregraduate patient safety education has attracted attention in recent years, with the introduction of pregraduate patient safety education in the Model Core Curriculum for Medical Education in 2001 and a significant increase in the description of patient safety in the Model Core Curriculum for Medical Education in 2022.”
Thank you for your valuable feedback.
We have made the suggested revisions. These changes have been incorporated into the revised manuscript’s Introduction Section, with adjustments on page 2, lines 58 to 62.
- Reviewer 2 commented “Please be aware of the verb past tense in the sentences. For example, in the introduction: “and compare it with” should be in the past tense”.
Thank you for your valuable feedback.
We have changed the wording to the past tense as suggested.
- Reviewer 2 commented “Some of the sentences are awkward or lack logical flow. For example: “Medical students were provided survey forms in April 4th, 2019, immediately after their enrollment. Furthermore, third year high school students were provided the forms in April 9th, 2019, immediately after progressing to the third year. The questionnaires were collected on the same day after all questions were answered.” How about something like: “The survey was carried out in April 2019, shortly after beginning the new school year.”
Thank you for your valuable feedback.
We have revised the wording to “The survey was carried out in April 2019, shortly after beginning the new school year.” This change has been incorporated into the revised manuscript’s Materials and Methods Section, with adjustments on page 2, lines 91 to 92.
- Reviewer 2 commented “In section 2.2 you mention: “The Cronbach’s alpha coefficient was 0.77.” However, at this stage it is not clear at this point if the is the Cronbach’s alpha for Nabilou et al.’s survey or for your Japanese version. Please clarify for the reader”.
Thank you for your valuable feedback.
The Cronbach’s alpha value (α=0.77) mentioned in Section 2.2 reflects the reliability of the Japanese version of the questionnaire. This information has been incorporated into the revised manuscript’s Materials and Methods Section, with adjustments on page 3, line 97.
- Reviewer 2 commented “Here I think it would be good to include (in parentheses)
information regarding the meanings of the terms, or content of the questions as you did in the preceding sentence regarding background: “Questions 5–15 asked about “perception,” questions 16–25 asked about “knowledge,” and questions 26–33 asked about “attitude.”
Thank you for your valuable feedback.
The methods section is indeed described in that manner.
- Reviewer 2 commented “Please be aware of the changes of voice, for example in the Materials and Methods section: “We calculated the total scores” should be in the passive voice to match the rest of that section.”
Thank you for your valuable feedback.
We have revised the wording to “the total scores were calculated.” This change has been incorporated into the revised manuscript’s Materials and Methods Section, with adjustments on page 3, line 109.
- Reviewer 2 commented “In the results section, you refer to the items in the
questionnaire as “questions”; however, they are written as statements. Maybe it would be better to refer to them as “items.” I think you can also remove the word “Question” from the tables and just have the numbers.”
Thank you for your valuable feedback.
We have changed “questions” to “items” throughout the manuscript and modified the tables to display only the question numbers instead of “Question.” These revisions have been incorporated into the revised manuscript’s Results Section.
- Reviewer 2 commented “Please use the word “participants” rather than “subjects”
Thank you for your valuable feedback.
We have changed “participants” to “subjects” throughout the manuscript. This revision has been incorporated into the revised manuscript’s Materials and Methods Section, with adjustments on page 2, line 83.
We hope this clarifies our approach and the rationale behind our research design. Again, thank you for your valuable time and efforts in reviewing our work.
Sincerely yours,

Reviewer 3 Report
Comments and Suggestions for Authors
To authors,
While the study compared two groups' perceptions of patient safety, it suffers from methodological errors and interpretation of results.
1) The abstract should state that the cohort of medical students studied were freshmen at the time of data collection.
2) Materials and Methods: Since the patient safety awareness scale was analyzed by dividing it into perception, knowledge, and attitude, the reliability of the tool should be presented not only in terms of overall reliability but also in terms of subdomain reliability.
3) Materials and Methods: We describe the process of translating the original tool in more detail, and the process of confirming expert validity.
4) Results: Because simple mean values are presented for each item on the scale, it is difficult to conclude that medical students actually have a higher mean for each survey item than high school students. You need to present statistical validation to show that the actual mean values are different. It would also be meaningful to present the mean values for each subscale combined and compare the two groups. Please revise Table 2~4.
5) Results: I think the main result of this study is the difference in total scores through the analysis of covariance. It would be better to present the statistics in a separate table rather than presenting the table as part of the figure.
6) Discussion: It would be nice if the discussion section drew out the implications of the results rather than repeating the results description.
7) Conclusions: You are missing a conclusion that should emphasize the significance of the study.
That's all for now.
Yours sincerely, Reviewer
Author Response
Dear Reviewer 3,
Thank you very much for your valuable feedback. We deeply appreciate your insightful comments.
1)Reviewer 3 commented “The abstract should state that the cohort of medical students studied were freshmen at the time of data collection.”
Thank you for your valuable feedback.
We have added the statement "Medical students were freshmen at the time of data collection" to the main text for clarity. This addition has been incorporated into the revised manuscript’s Abstract Section, with adjustments on page 1, line 24 to 25.
2)Reviewer 3 commented “Materials and Methods: Since the patient safety awareness scale was analyzed by dividing it into perception, knowledge, and attitude, the reliability of the tool should be presented not only in terms of overall reliability but also in terms of subdomain reliability.”
Thank you for your valuable feedback.
This study was conducted in accordance with the prior research by Nabilou et al. [1]. In their study, patient safety awareness was evaluated using the questionnaire as a whole, and reliability was assessed for the entire set of questions (α=0.72). Therefore, from the planning stages, this study did not intend to evaluate reliability for each subsection of "perception," "knowledge," and "attitude," and these values are not reported in the text.
References
- Nabilou B, Feizi A, Seyedin H. Patient Safety in Medical Education: Students' Perceptions, Knowledge and Attitudes. PLoS One. 2015 Aug 31;10(8):e0135610. doi: 10.1371/journal.pone.0135610. PMID: 26322897; PMCID: PMC4554725.
3)Reviewer 3 commented “Materials and Methods: We describe the process of translating the original tool in more detail, and the process of confirming expert validity.”
Thank you for your valuable feedback.
The questionnaire content was adapted from the previous study by Nabilou et al. [1] and translated through collaboration between English language specialists and patient safety experts. Subsequently, patient safety experts made minor modifications to reflect the current state of patient safety in Japan and its educational characteristics. A pilot study was conducted prior to this research to confirm the validity of the questionnaire, and reliability was also confirmed with a Cronbach’s alpha of 0.75.
This revision has been incorporated into the revised manuscript’s Materials and Methods Section, with adjustments on page 3, line 114 to 125.
References
- Nabilou B, Feizi A, Seyedin H. Patient Safety in Medical Education: Students' Perceptions, Knowledge and Attitudes. PLoS One. 2015 Aug 31;10(8):e0135610. doi: 10.1371/journal.pone.0135610. PMID: 26322897; PMCID: PMC4554725.
4)Reviewer 3 commented “Results: Because simple mean values are presented for each item on the scale, it is difficult to conclude that medical students actually have a higher mean for each survey item than high school students. You need to present statistical validation to show that the actual mean values are different. It would also be meaningful to present the mean values for each subscale combined and compare the two groups. Please revise Table 2~4.”
Thank you for your valuable feedback.
Firstly, to prevent inflation of the alpha error, we intentionally avoided conducting multiple tests on the results of each questionnaire item. This study was conducted in line with prior research by Nabilou et al. [1] and its predecessors, Madigosky et al. [2] and Leung et al. [3]. In these studies, patient safety awareness was assessed using the questionnaire as a whole; similarly, we evaluated patient safety awareness using the entire questionnaire. From the planning stage, we did not intend to calculate section-specific scores, and therefore, did not report them.
References
- Nabilou B, Feizi A, Seyedin H. Patient Safety in Medical Education: Students' Perceptions, Knowledge and Attitudes. PLoS One. 2015 Aug 31;10(8):e0135610. doi: 10.1371/journal.pone.0135610. PMID: 26322897; PMCID: PMC4554725.
- Madigosky WS, Headrick LA, Nelson K, Cox KR, Anderson T. Changing and sustaining medical students' knowledge, skills, and attitudes about patient safety and medical fallibility. Acad Med. 2006 Jan;81(1):94-101. doi: 10.1097/00001888-200601000-00022. PMID: 16377828.
- Leung GK, Patil NG. Patient safety in the undergraduate curriculum: medical students' perception. Hong Kong Med J. 2010 Apr;16(2):101-5. PMID: 20354243.
5)Reviewer 3 commented “Results: I think the main result of this study is the difference in total scores through the analysis of covariance. It would be better to present the statistics in a separate table rather than presenting the table as part of the figure.”
Thank you for your valuable feedback.
We have separated the figures and tables, adding a new graph of total scores comparing medical school freshmen and high school seniors as Figure 1, along with a new table as Table 5. These additions have been incorporated into the revised manuscript’s Results Section, with adjustments on page 7, line 236 to page 8, line 257.
6)Reviewer 3 commented “Discussion: It would be nice if the discussion section drew out the implications of the results rather than repeating the results description.”
Thank you for your valuable feedback.
We have restructured the Discussion Section. The results of this study suggest that medical school freshmen exhibit sensitivity not only to the technical aspects of medicine but also to patient safety from the outset of their medical education, indicating that their professional interests may already be well-formed in various areas. Furthermore, our findings imply that these freshmen are likely to actively engage in patient safety initiatives during their university education. Given their heightened awareness of patient safety, leveraging and enhancing this awareness could lead to more effective patient safety education.
In Japan, patient safety education is typically introduced in the third or fourth year of medical school. However, studies [1-4] report that student motivation in Japan and other countries tends to decline as they progress through school. Based on our findings, we recommend implementing patient safety education at an earlier stage, including more specialized content. These additions have been incorporated into the revised manuscript’s Discussion Section, with adjustments on page 8, line 259 to page 9, line 310.
In response to your suggestions, we have also added a Limitation Section to clarify the study’s limitations and future directions. This study has certain limitations. This study is a cross-sectional study comparing 132 medical students who enrolled at Teikyo University in 2019 with 166 third-year high school students from Teikyo University Junior and Senior High Schools. Further validation is necessary with medical freshmen from other universities in Japan, as well as with students from other countries with diverse educational and cultural backgrounds. Additionally, studies conducted in different years would strengthen these findings.
While this study suggests the effectiveness of earlier patient safety education, further research could confirm these results by comparing patient safety awareness post-education or at graduation between cohorts receiving early versus current timing of patient safety training. These additions have been incorporated under a new subheading in the revised manuscript’s Discussion Section, with adjustments on page 9, line 311 to 321.
References
- Nakashima et al. Surveys to assess the attitudes of medical students about learning. igakukiyouiku 2010,41(6):429-434. doi: https://doi.org/10.11307/mededjapan.41.429.
- Wu H, Li S, Zheng J, Guo J. Medical students' motivation and academic performance: the mediating roles of self-efficacy and learning engagement. Med Educ Online. 2020 Dec;25(1):1742964. doi: 10.1080/10872981.2020.1742964. PMID: 32180537; PMCID: PMC7144307.
- Sarkis AS, Hallit S, Hajj A, Kechichian A, Karam Sarkis D, Sarkis A, Nasser Ayoub E. Lebanese students' motivation in medical school: does it change throughout the years? A cross-sectional study. BMC Med Educ. 2020 Mar 31;20(1):94. doi: 10.1186/s12909-020-02011-w. PMID: 32234030; PMCID: PMC7110720.
- Kim KJ, Jang HW. Changes in medical students' motivation and self-regulated learning: a preliminary study. Int J Med Educ. 2015 Dec 28;6:213-5. doi: 10.5116/ijme.565e.0f87. PMID: 26708325; PMCID: PMC4695391.
7)Reviewer 3 commented “Conclusions: You are missing a conclusion that should emphasize the significance of the study.”
Thank you for your valuable feedback.
This study revealed that medical school freshmen possess a higher awareness of patient safety compared to high school seniors who do not aim for medical careers. This finding suggests that medical students already have a sensitivity to patient safety upon entry to medical school. Introducing patient safety education earlier in the medical curriculum could further enhance the awareness of medical students—and eventually physicians—contributing to the establishment of safer healthcare practices. Based on your suggestions, we have added a Conclusion Section to the revised manuscript and made adjustments on page 9, line 322 to page 10, line 328.

Reviewer 4 Report
Comments and Suggestions for Authors
Thank you for the opportunity to review this article, which is relevant to the professional practice of future physicians and the entire professional community in the health sector.
In general, the manuscript is easy to read and contains the necessary elements for methodological clarity. Notwithstanding the above, I believe it is necessary to improve/reformulate the section on discussion, bringing a real discussion of results, which is not the case in the current draft. Similarly, I believe it is necessary to separate the limitations subsection (within the discussion section) and point out avenues for future research and action on the topic studied.
A conclusion section should be incorporated.
Author Response
Dear Reviewer 4,
Thank you very much for your valuable feedback. We deeply appreciate your insightful comments.
1)Reviewer 4 commented “In general, the manuscript is easy to read and contains the necessary elements for methodological clarity. Notwithstanding the above, I believe it is necessary to improve/reformulate the section on discussion, bringing a real discussion of results, which is not the case in the current draft.”
Thank you for your valuable feedback.
We have restructured the Discussion Section. The results of this study suggest that medical school freshmen exhibit sensitivity not only to the technical aspects of medicine but also to patient safety from the outset of their medical education, indicating that their professional interests may already be well-formed in various areas. Furthermore, our findings imply that these freshmen are likely to actively engage in patient safety initiatives during their university education. Given their heightened awareness of patient safety, leveraging and enhancing this awareness could lead to more effective patient safety education.
In Japan, patient safety education is typically introduced in the third or fourth year of medical school. However, studies [1-4] report that student motivation in Japan and other countries tends to decline as they progress through school. Based on our findings, we recommend implementing patient safety education at an earlier stage, including more specialized content.
These additions have been incorporated into the revised manuscript’s Discussion Section, with adjustments on page 8, line 258 to page 9, line 309.
References
- Nakashima et al. Surveys to assess the attitudes of medical students about learning. igakukiyouiku 2010,41(6):429-434. doi: https://doi.org/10.11307/mededjapan.41.429.
- Wu H, Li S, Zheng J, Guo J. Medical students' motivation and academic performance: the mediating roles of self-efficacy and learning engagement. Med Educ Online. 2020 Dec;25(1):1742964. doi: 10.1080/10872981.2020.1742964. PMID: 32180537; PMCID: PMC7144307.
- Sarkis AS, Hallit S, Hajj A, Kechichian A, Karam Sarkis D, Sarkis A, Nasser Ayoub E. Lebanese students' motivation in medical school: does it change throughout the years? A cross-sectional study. BMC Med Educ. 2020 Mar 31;20(1):94. doi: 10.1186/s12909-020-02011-w. PMID: 32234030; PMCID: PMC7110720.
- Kim KJ, Jang HW. Changes in medical students' motivation and self-regulated learning: a preliminary study. Int J Med Educ. 2015 Dec 28;6:213-5. doi: 10.5116/ijme.565e.0f87. PMID: 26708325; PMCID: PMC4695391.
2)Reviewer 4 commented “Similarly, I believe it is necessary to separate the limitations subsection (within the discussion section) and point out avenues for future research and action on the topic studied.”
Thank you for your valuable feedback.
We have added a Limitation Section to clarify the study's limitations and future directions. This study has certain limitations. This study is a cross-sectional study comparing 132 medical students who enrolled at Teikyo University in 2019 with 166 third-year high school students from Teikyo University Junior and Senior High Schools. Further validation is necessary with medical freshmen from other universities in Japan, as well as with students from other countries with diverse educational and cultural backgrounds. Additionally, studies conducted in different years would strengthen these findings.
While this study suggests the effectiveness of earlier patient safety education, further research could confirm these results by comparing patient safety awareness post-education or at graduation between cohorts receiving early versus current timing of patient safety training. Based on your suggestions, we have incorporated these additions under a new subheading in the revised manuscript’s Discussion Section, with adjustments page 9, line 312 to line 322.
3)Reviewer 4 commented “A conclusion section should be incorporated.”
Thank you for your valuable feedback.
This study revealed that medical school freshmen possess a higher awareness of patient safety compared to high school seniors who do not aim for medical careers. This finding suggests that medical students already have a sensitivity to patient safety upon entry to medical school. Introducing patient safety education earlier in the medical curriculum could further enhance the awareness of medical students—and eventually physicians—contributing to the establishment of safer healthcare practices. Based on your suggestions, we have added a Conclusion Section to the revised manuscript and made adjustments on page 9, line 323 to page 10, line 329.

Reviewer 5 Report
Comments and Suggestions for Authors
Thank you for the opportunity to review this manuscript. This work explores the important issue of patient safety awareness among two distinct groups - medical school freshmen and high school seniors. The authors make a strong case for the need for patient safety education from an early stage in medical training. Detailed comments are provided below for the various sections of the manuscript, for the authors to improve their paper.
Introduction: the paper could benefit from further explanation of why high school seniors were chosen as a control group and how their awareness might reflect the general public's understanding. The justification for this comparison is somewhat implied but not explicitly addressed, potentially weakening the argument.
Methodology: the study employs a self-administered, anonymous questionnaire divided into three categories: perception, knowledge, and attitude. This division provides a comprehensive framework for understanding patient safety awareness. The Cronbach’s alpha coefficient of 0.77 suggests a reliable measure.
While the methodology is sound in terms of its statistical rigor, the paper lacks a detailed discussion on how/if the survey questions were tailored to assess the specific nuances of patient safety awareness among such different groups. For instance, medical students, even at admission, may have a natural inclination toward learning about medical safety, which could skew the results. A more detailed explanation/a comment of how/if the questionnaire mitigates such biases would strengthen the methodological section.
Results: The paper highlights a clear difference between medical students and high school seniors, with medical students scoring significantly higher in both the “perception” and “attitude” categories. The use of a Likert scale is appropriate for capturing attitudes and perceptions, and the statistical significance of the results is well-supported through the use of student’s t-tests and ANCOVA.
One point of critique is the lack of deeper analysis of the “knowledge” component, which showed low positive responses from both groups. The authors acknowledge that neither group had a high level of knowledge about patient safety, but this finding could have been explored further. What factors contribute to this gap? Could there be external influences or is this purely a result of educational shortcomings?
Discussion: The discussion rightly points out that medical students, despite being new to medical education, have a stronger perception and attitude toward patient safety. The paper supports the idea of integrating patient safety education early in medical training, which is an important implication.
However, the paper misses an opportunity to address the practical challenges of implementing patient safety education in schools. How feasible is it to introduce comprehensive safety training at the high school level? Moreover, the discussion could explore the broader implications for medical curricula, beyond just emphasizing the need for early education. Title for this section should be "Discussion and conclusions" if the authors wish to keep them together, otherwise it seems like a section is missing.
Comments on the Quality of English LanguageMinor/moderate editing of English language required.
Author Response
Dear Reviewer 5,
Thank you very much for your valuable feedback. We deeply appreciate your insightful comments.
1)Reviewer 5 commented “Introduction: the paper could benefit from further explanation of why high school seniors were chosen as a control group and how their awareness might reflect the general public's understanding. The justification for this comparison is somewhat implied but not explicitly addressed, potentially weakening the argument.”
Thank you for your valuable feedback.
In Japan, most medical students enter medical school directly after high school. The subjects of this study are medical school freshmen who have just enrolled, prior to the start of formal medical education. Therefore, we selected high school seniors as the control group, as they are closest in age and have a similar educational level. The only difference between them and our comparison group—third-year high school students—is whether or not they have applied to medical school. Based on your suggestions, we have incorporated these points into the revised manuscript's Materials and Methods Section, with adjustments on page 2, line 83 to line 92.
2)Reviewer 5 commented “Methodology: the study employs a self-administered, anonymous questionnaire divided into three categories: perception, knowledge, and attitude. This division provides a comprehensive framework for understanding patient safety awareness. The Cronbach’s alpha coefficient of 0.77 suggests a reliable measure.
While the methodology is sound in terms of its statistical rigor, the paper lacks a detailed discussion on how/if the survey questions were tailored to assess the specific nuances of patient safety awareness among such different groups. For instance, medical students, even at admission, may have a natural inclination toward learning about medical safety, which could skew the results. A more detailed explanation/a comment of how/if the questionnaire mitigates such biases would strengthen the methodological section.”
Thank you for your valuable feedback.
The survey content used in this study was adapted from previous research, translated into Japanese through collaboration with English language specialists and patient safety experts, and modified to incorporate aspects specific to medical safety education in Japan. Prior to this study, a pilot survey was conducted to confirm the validity of the questionnaire, and Cronbach’s alpha was employed to ensure reliability. The survey used in this study primarily included objective questions, excluding ambiguous or subjective elements. Additionally, any potential biases related to age and gender were adjusted statistically using ANCOVA. These details are addressed in the Method Section on page 3, line 114 to 125.
3)Reviewer 5 commented “Results: The paper highlights a clear difference between medical students and high school seniors, with medical students scoring significantly higher in both the “perception” and “attitude” categories. The use of a Likert scale is appropriate for capturing attitudes and perceptions, and the statistical significance of the results is well-supported through the use of student’s t-tests and ANCOVA.
One point of critique is the lack of deeper analysis of the “knowledge” component, which showed low positive responses from both groups. The authors acknowledge that neither group had a high level of knowledge about patient safety, but this finding could have been explored further. What factors contribute to this gap? Could there be external influences or is this purely a result of educational shortcomings?”
Thank you for your valuable feedback.
The results indicated that medical school freshmen demonstrated greater knowledge of commonly understood concepts such as team medicine (item 21) and informed consent (item 23) compared to high school seniors. Additionally, although their level of specialized knowledge was not particularly high, as you noted, a higher proportion of medical school freshmen reported familiarity with specialized concepts than high school seniors. This suggests that medical school freshmen may have more knowledge about patient safety—even in areas they may not find particularly interesting—compared to non-medical students, suggesting a broader interest in various aspects of medicine. Based on your suggestions, we have added this discussion to the revised manuscript’s Discussion Section and made adjustments on page 9, line 286 to 301.
4)Reviewer 5 commented “Discussion: The discussion rightly points out that medical students, despite being new to medical education, have a stronger perception and attitude toward patient safety. The paper supports the idea of integrating patient safety education early in medical training, which is an important implication.
However, the paper misses an opportunity to address the practical challenges of implementing patient safety education in schools. How feasible is it to introduce comprehensive safety training at the high school level? Moreover, the discussion could explore the broader implications for medical curricula, beyond just emphasizing the need for early education. Title for this section should be "Discussion and conclusions" if the authors wish to keep them together, otherwise it seems like a section is missing.”
Thank you for your valuable feedback.
We have restructured the Discussion Section. The results of this study suggest that medical school freshmen exhibit sensitivity not only to the technical aspects of medicine but also to patient safety from the outset of their medical education, indicating that their professional interests may already be well-formed in various areas. Furthermore, our findings imply that these freshmen are likely to actively engage in patient safety initiatives during their university education. Given their heightened awareness of patient safety, leveraging and enhancing this awareness could lead to more effective patient safety education.
In Japan, patient safety education is typically introduced in the third or fourth year of medical school. However, studies [1-4] report that student motivation in Japan and other countries tends to decline as they progress through school. Based on our findings, we recommend implementing patient safety education at an earlier stage, including more specialized content. These additions have been incorporated into the revised manuscript’s Discussion Section, with adjustments on page 8, line 259 to page 9, line 311.
In response to your suggestions, we have also added a Conclusion Section.
This study revealed that medical school freshmen possess a higher awareness of patient safety compared to high school seniors who do not aim for medical careers. This finding suggests that medical students already have a sensitivity to patient safety upon entry to medical school. Introducing patient safety education earlier in the medical curriculum could further enhance the awareness of medical students—and eventually physicians—contributing to the establishment of safer healthcare practices. Based on your suggestions, we have added a Conclusion Section to the revised manuscript and made adjustments on page 9, line 323 to page 10, line 329.
References
- Nakashima et al. Surveys to assess the attitudes of medical students about learning. igakukiyouiku 2010,41(6):429-434. doi: https://doi.org/10.11307/mededjapan.41.429.
- Wu H, Li S, Zheng J, Guo J. Medical students' motivation and academic performance: the mediating roles of self-efficacy and learning engagement. Med Educ Online. 2020 Dec;25(1):1742964. doi: 10.1080/10872981.2020.1742964. PMID: 32180537; PMCID: PMC7144307.
- Sarkis AS, Hallit S, Hajj A, Kechichian A, Karam Sarkis D, Sarkis A, Nasser Ayoub E. Lebanese students' motivation in medical school: does it change throughout the years? A cross-sectional study. BMC Med Educ. 2020 Mar 31;20(1):94. doi: 10.1186/s12909-020-02011-w. PMID: 32234030; PMCID: PMC7110720.
- Kim KJ, Jang HW. Changes in medical students' motivation and self-regulated learning: a preliminary study. Int J Med Educ. 2015 Dec 28;6:213-5. doi: 10.5116/ijme.565e.0f87. PMID: 26708325; PMCID: PMC4695391.
Round 2
Reviewer 1 Report
Comments and Suggestions for Authors
The authors tried to improve the quality of the manuscript. However, they committed the same errors. In fact, the idea need to be readapted for only medical students. The sentence "most of medical students enter medical school directly after high school" means that some of them have another way to enroll in medical studies. In addition, the items from 8 to 15 (perception) and 16-25 (knowledge) are adapted for healthcare workers or an "experienced" student but not for a first year student (and surely for non medical students). We cannot ask a student a question like " If I saw a medical error, I would report it to my supervisor…" he has not yet formed for this technical subject and he has not yet a supervisor.
You should readapt your results just for the non technical terms and items.
The authors should also, delete the term "general population" (from the tile, main text and tables) and readapt the title accordingly.
They should also use statistical analysis in the comparison of each item.
Delete the table S1 and add it as supplementary materials
Author Response
Dear Reviewer 1,
Thank you very much for your valuable feedback. We deeply appreciate your insightful comments.
1)Reviewer 1 commented “The authors tried to improve the quality of the manuscript. However, they committed the same errors. In fact, the idea need to be readapted for only medical students. The sentence "most of medical students enter medical school directly after high school" means that some of them have another way to enroll in medical studies.”
Thank you for your valuable comments.
In Japan, it is common for students to proceed directly to medical school after graduating from high school. However, admission to medical school is very competitive, which can sometimes result in delays before successfully gaining entry. There are no alternative pathways for admission; all applicants must take the same entrance examination for medical school.
2)Reviewer 1 commented “In addition, the items from 8 to 15 (perception) and 16-25 (knowledge) are adapted for healthcare workers or an "experienced" student but not for a first year student (and surely for non medical students). We cannot ask a student a question like " If I saw a medical error, I would report it to my supervisor…" he has not yet formed for this technical subject and he has not yet a supervisor. You should readapt your results just for the non technical terms and items.”
Thank you for your valuable comments.
Previous studies have reported that a certain number of incoming medical students exhibit a high level of awareness regarding patient safety [1,2], suggesting that these students develop interests in diverse areas, including patient safety, beyond the technical aspects of medicine. This study aims to compare the awareness of patient safety—an area that may initially be less engaging for medical students due to its non-technical nature—between medical students and non-medical students. It should be noted that this study is not designed to measure terminology comprehension but rather to compare the perspectives of both groups.
This study followed the prior research conducted by Nabilou et al. [3] and its predecessors, Madigosky et al. [4] and Leung et al. [5]. In these previous studies [3-5], Cronbach’s alpha was measured for the entire questionnaire, and patient safety awareness was then evaluated; we adopted the same approach. We had not initially planned to extract and analyze only the non-technical items. However, based on your suggestion, we conducted an analysis focusing on the "attitude" items, which represent non-technical aspects. Specifically, we calculated the total scores and presented them as the mean ± standard deviation along with the 95% confidence intervals (CI). The results showed that medical students scored 34.0 ± 3.0 and controls scored 30.1 ± 3.6 (mean difference: 3.3, p < 0.001, 95% CI: 2.5 - 4.2). ANCOVA also demonstrated a statistically significant difference between medical students and controls (Least Square mean: 2.9, p < 0.001, 95% CI: 1.7 - 4.0). These findings suggest that medical students scored higher than controls on non-technical items, indicating a greater awareness of patient safety.
Based on your suggestions, we have incorporated these points into the revised manuscript's Result Section, with adjustments on page 7, line 222 to 225.
References
- Kasai K, Shu A, Otaki Y. A study of medical safety education for students at the time of admission. Health Prof Educ. 2016/2017;2 & 3: 21-28.
- Shu A, Kasai K, Sasamori C, Hase S, Otaki Y. An analysis of patient safety awareness of first-year medical students. Health Prof Educ. 2018;4: 17-21.
- Nabilou B, Feizi A, Seyedin H. Patient Safety in Medical Education: Students' Perceptions, Knowledge and Attitudes. PLoS One. 2015 Aug 31;10(8):e0135610. doi: 10.1371/journal.pone.0135610. PMID: 26322897; PMCID: PMC4554725.
- Madigosky WS, Headrick LA, Nelson K, Cox KR, Anderson T. Changing and sustaining medical students' knowledge, skills, and attitudes about patient safety and medical fallibility. Acad Med. 2006 Jan;81(1):94-101. doi: 10.1097/00001888-200601000-00022. PMID: 16377828.
- Leung GK, Patil NG. Patient safety in the undergraduate curriculum: medical students' perception. Hong Kong Med J. 2010 Apr;16(2):101-5. PMID: 20354243.
3)Reviewer 1 commented “The authors should also, delete the term "general population" (from the tile, main text and tables) and readapt the title accordingly.”
Thank you for your valuable comments.
Following the suggestion of other reviewers, we initially changed the term for the control group to “general public.” However, in accordance with Reviewer 1’s constructive feedback, we have revised this designation to “Controls.”
4)Reviewer 1 commented “They should also use statistical analysis in the comparison of each item.”
Thank you for your valuable comments.
To avoid increasing the risk of Type I error, we did not perform multiple testing for the results of each survey item. As you suggested, we calculated the total score for the non-technical “attitude” items and presented the mean ± standard deviation along with the 95% confidence intervals (CI). The results showed that medical students scored 34.0 ± 3.0 and controls scored 30.1 ± 3.6 (mean difference: 3.3, p < 0.001, 95% CI: 2.5 - 4.2). ANCOVA also demonstrated a statistically significant difference between medical students and controls (Least Square mean: 2.9, p < 0.001, 95% CI: 1.7 - 4.0).
5)Reviewer 1 commented “Delete the table S1 and add it as supplementary materials”
Thank you for your valuable comments.
We have removed Table S1 from the main text and included it in the Supplementary Materials.

Reviewer 2 Report
Comments and Suggestions for Authors
A Comparative Study of Patient Safety Awareness Among Medical School Freshmen and the General Public of the Same Age Group
While I think the authors have made a good effort to address the points raised by the reviewers, the manuscript needs further work to improve it, mainly related to English grammar, before it is accepted for publication. In many instances the revised or additional sections have added in more mistakes and confusion. I highly suggest using a professional editor.
• Line 77: You state that part of the purpose of the study is to “explore the introduction and optimal timing of patient safety education in medical education.” However, from your study design you cannot really do that.” Maybe you should be more subtle here, something like “discuss the introduction and timing of patient safety education in medical education.” The same issue is encountered in and 262-264.
• I think the authors have answered my comment about the control group, however, I would suggest referring to the group as aged-matched controls. “Aged-matched individuals who were not medical students were recruited as a control group.” In the manuscript and Tables, you can refer to them as ‘medical students’ and ‘controls’, and ‘medical student group’ and ‘control group.’ I think this would be clearer.
• With regards mentioning the fact that they are from “Teikyo University Junior and Senior High School” I suggest that you can simply state that they are from “an affiliated institution within the Teikyo University organization.” This way you could avoid having to explain about the education level of high school seniors versus freshmen medical students etc. (Lines 72-78, Lines 83-92, 314, etc.)
• Suggested title: A Comparative Study on Patient Safety Awareness Between Medical School Freshmen and Age-Matched Individuals
• I made the comment that you should refer not to study “subjects” but to “participants.” This is in-line with the guidance of the AMA manual of style. You responded that you had done this, however, you consistently use the word subjects in the “manuscript”. I would suggest changing it. (Lines 81, 83, 126 etc.)
• In the results section, try to add some quick interpretation of the results. For example (Lines 255-257): “ANCOVA was also performed on the total scores, and a statistically significant difference was observed between the two groups (Least Square mean: 13.1, p < 0.001, 95% CI: 9.7 - 16.5), demonstration that medical students have greater… This is not an exhaustive discussion, just a quick interpretation.
• In the discussion section I think it would be good if you could contextualize your findings and include some information about published studies regarding patient safety education. You state in the introduction “many studies measuring patient safety awareness among medical undergraduates have been conducted in various countries [17-26]”. Maybe you can tell us about what these studies have to say about patient safely education as it pertains to your study findings.
• The conclusion of the study is very weak. You state “This study revealed that medical school freshmen possess a higher awareness of patient safety compared to the general public of the same age group who do not aim for medical careers.” This finding is too obvious and lacks any sort of subtle or interesting statement. It is like saying “We found that students who entered art school are better at drawing than students who did not.” In the conclusion, highlight the importance of patient safety and education in this field. Then please restate your research question and/or hypothesis. Restate the main findings of your study and mention the main implications of your findings for medical school educational practice.
As mentioned above, one of the main problems is the English grammar, especially in the revised sections. The manuscript should be properly edited by an experienced editor in the field, preferably a native English speaker. Below I will list some points I picked up on, but this is not a complete list.
• Lines 83 onwards: These two sections of texts are basically repeating the same information, so you should choose one or the other: “The subjects of this study were medical school freshmen who, although enrolled in medical school, had not yet received any formal medical education. As a comparison group, we selected third-year high school students.” AND “The subjects of this study are medical school freshmen who have just enrolled, prior to the start of formal medical education. Therefore, we selected high school seniors as the control group, as they are closest in age and have a similar educational level.”
• It is the same with lines 240-243 and lines 255-258.
• Sections 114-122. Please check the logical flow of the sentences.
• Lines 240-242: Check grammar: “After controlling for confounders through ANCOVA were shown in Table5, a statistically significant difference was observed in the total scores between the two groups as well.”
• The title of Table 5 doesn’t give the reader enough information to interpret the contents.
• Lines 276, 286: The following phrase is repeated and from an English perspective it sounds very strange to the reader because of course medical students receive medical education “medical students who received medical education are more aware of the need to report medical errors” “even medical students who received medical education”
• Line 279: “Conversely” is the wrong word here are you are not providing contrary information.
• There are numerous other issues relate to grammar and flow of sentences, redundancy, repetition, etc. Thus, I strongly recommend a professional editor.
Comments on the Quality of English Language
As stated above, a thorough, professional edit of this manuscript is hight recommended. There are many issues related to grammar, flow of ideas and sentences, paragraph structure, etc.
Author Response
Dear Reviewer 2,
Thank you very much for your valuable feedback. We deeply appreciate your insightful comments.
1)Reviewer 2 commented “Line 77: You state that part of the purpose of the study is to “explore the introduction and optimal timing of patient safety education in medical education.” However, from your study design you cannot really do that.” Maybe you should be more subtle here, something like “discuss the introduction and timing of patient safety education in medical education.” The same issue is encountered in and 262-264.”
Thank you for your valuable comments.
We have made the revisions in accordance with your suggestions. We have incorporated these points into the revised manuscript's Materials and Methods Section and Discussion Section, with adjustments on page 2, line 75 and page 8, line 258.
2)Reviewer 2 commented “I think the authors have answered my comment about the control group, however, I would suggest referring to the group as aged-matched controls. “Aged-matched individuals who were not medical students were recruited as a control group.” In the manuscript and Tables, you can refer to them as ‘medical students’ and ‘controls’, and ‘medical student group’ and ‘control group.’ I think this would be clearer.”
Thank you for your valuable comments.
In accordance with your suggestions, we have revised the text as indicated. Additionally, we have updated the terminology in the manuscript and tables to refer to “medical students” and “controls.”
3)Reviewer 2 commented “With regards mentioning the fact that they are from “Teikyo University Junior and Senior High School” I suggest that you can simply state that they are from “an affiliated institution within the Teikyo University organization.” This way you could avoid having to explain about the education level of high school seniors versus freshmen medical students etc. (Lines 72-78, Lines 83-92, 314, etc.)”
Thank you for your valuable comments.
In accordance with your suggestions, we have updated the term to “an affiliated institution within the Teikyo University organization.” We have incorporated these points into the revised manuscript's Introduction Section and Materials and Methods Section, with adjustments on page 2, line 70 to 75 and page 2, line 78 to 83.
4)Reviewer 2 commented “Suggested title: A Comparative Study on Patient Safety Awareness Between Medical School Freshmen and Age-Matched Individuals”
Thank you for your valuable comments.
In accordance with your suggestion, we have revised the title to “A Comparative Study on Patient Safety Awareness Between Medical School Freshmen and Age-Matched Individuals.”
5)Reviewer 2 commented “ I made the comment that you should refer not to study “subjects” but to “participants.” This is in-line with the guidance of the AMA manual of style. You responded that you had done this, however, you consistently use the word subjects in the “manuscript”. I would suggest changing it. (Lines 81, 83, 126 etc.)”
Thank you for your valuable comments.
In accordance with your suggestion, we have changed the term “subjects” to “participants” throughout the manuscript.
6)Reviewer 2 commented “In the results section, try to add some quick interpretation of the results. For example (Lines 255-257): “ANCOVA was also performed on the total scores, and a statistically significant difference was observed between the two groups (Least Square mean: 13.1, p < 0.001, 95% CI: 9.7 - 16.5), demonstration that medical students have greater… This is not an exhaustive discussion, just a quick interpretation.”
Thank you very much for your valuable comments.
In accordance with your suggestion, we have included a brief interpretation. We have incorporated these points into the revised manuscript's Result Section, with adjustments on page 4, line 161 to 162 and page 7, line 239 to 240.
7)Reviewer 2 commented “In the discussion section I think it would be good if you could contextualize your findings and include some information about published studies regarding patient safety education. You state in the introduction “many studies measuring patient safety awareness among medical undergraduates have been conducted in various countries [17-26]”. Maybe you can tell us about what these studies have to say about patient safely education as it pertains to your study findings.”
Thank you very much for your valuable comments. We made changes in the texts as follows.
Previous studies [1-5] have reported that patient safety awareness tends to increase following patient safety education and that motivation to learn tends to decrease as student’s progress through the years. Our study revealed that medical school freshmen demonstrated a significantly higher motivation to learn about patient safety compared to age-matched controls. This suggests that medical school freshmen are interested not only in the technical aspects of medicine but also in patient safety. It is anticipated that implementing continuous patient safety education, including knowledge of patient safety, from the earliest post-enrollment period—when motivation to learn is at its highest—could yield even higher educational outcomes.
We have incorporated these points into the revised manuscript's Discussion Section, with adjustments on page 9, line 297 to 309.
References
- Montilla-Herrador J, Lozano-Meca JA, Baño-Alcaraz A, Lillo-Navarro C, Martín-San Agustín R, Gacto-Sánchez M. Knowledge and Attitudes towards Patient Safety among Students in Physical Therapy in Spain: A Longitudinal Study. Int J Environ Res Public Health. 2022 Sep 15;19(18):11618. doi: 10.3390/ijerph191811618. PMID: 36141888; PMCID: PMC9517046.
- Svitlica BB, Šajnović M, Simin D, Ivetić J, Milutinović D. Patient safety: Knowledge and attitudes of medical and nursing students: Cross-sectional study. Nurse Educ Pract. 2021 May;53:103089. doi: 10.1016/j.nepr.2021.103089. Epub 2021 May 18. PMID: 34049090.
- Nakashima et al. Surveys to assess the attitudes of medical students about learning. igakukiyouiku 2010,41(6):429-434. doi: https://doi.org/10.11307/mededjapan.41.429.
- Wu H, Li S, Zheng J, Guo J. Medical students' motivation and academic performance: the mediating roles of self-efficacy and learning engagement. Med Educ Online. 2020 Dec;25(1):1742964. doi: 10.1080/10872981.2020.1742964. PMID: 32180537; PMCID: PMC7144307.
- Sarkis AS, Hallit S, Hajj A, Kechichian A, Karam Sarkis D, Sarkis A, Nasser Ayoub E. Lebanese students' motivation in medical school: does it change throughout the years? A cross-sectional study. BMC Med Educ. 2020 Mar 31;20(1):94. doi: 10.1186/s12909-020-02011-w. PMID: 32234030; PMCID: PMC7110720.
8)Reviewer 2 commented “The conclusion of the study is very weak. You state “This study revealed that medical school freshmen possess a higher awareness of patient safety compared to the general public of the same age group who do not aim for medical careers.” This finding is too obvious and lacks any sort of subtle or interesting statement. It is like saying “We found that students who entered art school are better at drawing than students who did not.” In the conclusion, highlight the importance of patient safety and education in this field. Then please restate your research question and/or hypothesis. Restate the main findings of your study and mention the main implications of your findings for medical school educational practice.”
Thank you very much for your valuable comments.
We made changes in the texts as follows.
In Japan, approximately 300 medical accidents were reported to the Medical Accident Investigation System annually, indicating the importance of patient safety education. However, patient safety education curricula widely vary among medical schools and have not yet been standardized across the country. This study aimed to clarify the characteristics of patient safety awareness among medical school freshmen prior to receiving patient safety education by comparing them with age-matched individuals who are not pursuing a medical career. We also aimed to discuss the introduction of patient safety education in medical schools and the optimal timing for its implementation.
Our findings revealed that medical school freshmen demonstrated significantly higher motivation to learn about patient safety compared to their age-matched controls. This suggested that medical school freshmen are interested not only in the technical aspects of medicine but also in patient safety. In other words, medical school freshmen appear to have already developed professional interests at the time of admission, encompassing not only technical aspects of healthcare but also broader areas such as patient safety. Therefore, incorporating continuous patient safety education from the first year—when students’ motivation to learn is considered to be at its highest—may enhance medical students’ and, ultimately, physicians’ awareness of patient safety, contributing to the promotion of safer healthcare practices.
We have incorporated these points into the revised manuscript's Conclusion Section, with adjustments on page 9, line 322 to 340.
9)Reviewer 2 commented “As mentioned above, one of the main problems is the English grammar, especially in the revised sections. The manuscript should be properly edited by an experienced editor in the field, preferably a native English speaker.”
Thank you very much for your valuable comments.
Following your suggestions, the English in this manuscript was proofread by experts in patient safety as well as specialists in medical English.
10)Reviewer 2 commented “Lines 83 onwards: These two sections of texts are basically repeating the same information, so you should choose one or the other: “The subjects of this study were medical school freshmen who, although enrolled in medical school, had not yet received any formal medical education. As a comparison group, we selected third-year high school students.” AND “The subjects of this study are medical school freshmen who have just enrolled, prior to the start of formal medical education. Therefore, we selected high school seniors as the control group, as they are closest in age and have a similar educational level.”
Thank you very much for your valuable comments.
We have made changes to the texts as follows.
The participants of this study were 132 medical school freshmen at the time of admission in the academic year of 2019, who had not yet received formal medical education. The controls of this study were 166 high school seniors enrolled at the affiliated institution, as they were closest in age and had a similar educational level within the Japanese educational system, in which medical schools accept students as early as the last term of their high school senior year.
We have incorporated these points into the revised manuscript's Materials and Methods Section, with adjustments on page 2, line 78 to 83.
11)Reviewer 2 commented “It is the same with lines 240-243 and lines 255-258.”
Thank you very much for your valuable comments. We have made the corrections as suggested.
12)Reviewer 2 commented “Sections 114-122. Please check the logical flow of the sentences.”
Thank you very much for your valuable comments.
We have made the corrections as suggested. We have incorporated these points into the revised manuscript's Materials and Methods Section, with adjustments on page 3, line 106 to 117.
13)Reviewer 2 commented “Lines 240-242: Check grammar: “After controlling for confounders through ANCOVA were shown in Table5, a statistically significant difference was observed in the total scores between the two groups as well.””
Thank you very much for your valuable comments.
We have made the corrections as suggested. We have incorporated these points into the revised manuscript's Result Section, with adjustments on page 7, line 236 to 239.
14)Reviewer 2 commented “The title of Table 5 doesn’t give the reader enough information to interpret the contents.”
Thank you very much for your valuable comments.
We have made the corrections as suggested. We have incorporated these points into the revised manuscript's Result Section, with adjustments on page 8, line 249.
15)Reviewer 2 commented “Lines 276, 286: The following phrase is repeated and from an English perspective it sounds very strange to the reader because of course medical students receive medical education “medical students who received medical education are more aware of the need to report medical errors” “even medical students who received medical education””
Thank you very much for your valuable comments.
We have revised the expressions to “medical students who received patient safety education are more aware of the need to report medical errors” and “even medical students who received patient safety education.”
We have incorporated these points into the revised manuscript's Discussion Section, with adjustments on page 8, line 270 to 271 and page 9, line 280 to 281.
16)Reviewer 2 commented “Line 279: “Conversely” is the wrong word here are you are not providing contrary information.”
Thank you very much for your valuable comments. In accordance with your suggestion, we have removed “Conversely.”

Reviewer 3 Report
Comments and Suggestions for Authors
The submitted paper has been well revised, thank you for your efforts.
Author Response
Dear Reviewer 3,
Thank you very much for your valuable feedback. We deeply appreciate your insightful comments.
1)Reviewer 3 commented “The submitted paper has been well revised, thank you for your efforts. ”
Your constructive feedback has greatly helped us improve the manuscript. We sincerely appreciate the time and effort you dedicated to reviewing our work.
Reviewer 4 Report
Comments and Suggestions for Authors
The manuscript, in my opinion, in this second round of the revision process, has improved substantially, which is why, in my opinion, it should be accepted for publication. I only recommend that the table presented at the end of the manuscript as ‘supplementary material’ be removed and inserted in a separate file.
Author Response
Dear Reviewer 4,
Thank you very much for your valuable feedback. We deeply appreciate your insightful comments.
1)Reviewer 4 commented “The manuscript, in my opinion, in this second round of the revision process, has improved substantially, which is why, in my opinion, it should be accepted for publication. I only recommend that the table presented at the end of the manuscript as ‘supplementary material’ be removed and inserted in a separate file.”
Thank you for your valuable comments.
We have removed Table S1- S3 from the main text and included it in the Supplementary Materials. Your constructive feedback has greatly helped us improve the manuscript. We sincerely appreciate the time and effort you dedicated to reviewing our work.